# Sulfonated tryptanthrin anolyte increases performance in pH neutral aqueous redox flow batteries

Daniela Pinheiro[1], Marta Pineiro [1] & J. Sérgio Seixas de Melo [1]✉

Aqueous organic redox flow batteries (AORFBs) hold great promise as low-cost, environmentally friendly and safe alternative energy storage media. Here we present aqueous organometallic and all-organic active materials for RFBs with a water-soluble active material, sulfonated tryptanthrin (TRYP-SO$_3$H), working at a neutral pH and showing long-term stability. Electrochemical measurements show that TRYP-SO$_3$H displays reversible peaks at neutral pH values, allowing its use as an anolyte combined with potassium ferrocyanide or 4,5-dihydroxy-1,3-benzenedisulfonic acid disodium salt monohydrate as catholytes. Single cell tests show reproducible charge-discharge cycles for both catholytes, with significantly improved results for the aqueous all-organic RFB reaching high cell voltage (0.94 V) and high energy efficiencies, stabilized during at least 50 working cycles.

[1] University of Coimbra, CQC, Department of Chemistry, Rua Larga, Coimbra, Portugal. ✉email: sseixas@ci.uc.pt

The global and economic population growth impels the increase of energy consumption. Resources formed over hundreds of millions of years have been burned in a relatively short time, with substantial environmental impact[1,2]. To reduce the use of fossil fuels, and environmentally friendly route to generate and store electricity from renewable sources is needed to fulfill the world's needs in a sustainable way[2–7].

In the past recent years, renewable energy technologies have therefore attracted much scientific and public interest[8]. Traditional batteries such as lithium-ion batteries are of widespread use, but they cannot cost-effectively store, enough energy for the long discharge durations at rated power, they use flammable organic electrolytes and reveal high maintenance costs[8–10].

Redox flow batteries (RFBs) are an emerging and highly promising power source, representing one of the best storage technologies for electrical energy that is obtained from renewable sources like wind power and solar energy[11]. RFBs are frequently described as affordable, reliable (with extremely long charge/discharge cycle life) and eco-friendly depending on the materials used[5,12–15]. Aqueous organic redox flow batteries (AORFBs) have been recently proposed as low-cost and alternatives to metal-based RFBs technology[5,8–10,16–19]. These AORFBs have several outstanding advantages mainly because of their prospect of offering such a grid-scale energy storage solution[20]. In addition, AORFBs are more environmentally friendly and safe since they use nonflammable aqueous redox-active electrolytes[9,21].

In the last years, novel AORFBs designs have been proposed essentially based on different electrolytes[8–10,15,22,23]. The use of anthraquinones has gained popularity since these quinones are well-known redox-active molecules with electrochemical reversibility and fast reactions rates[9,16,24–27]. However, and in particular, at alkaline media, this family of compounds still seems rather unstable and most of the RFBs cannot meet requirements for practical application[8]. To overcome the limitation of the corrosive electrolytes (acidic or alkaline media), the development of neutral aqueous RFBs has emerged over the years[9,20,25,28,29]. Neutral AORFBs are more ecofriendly and have outstanding advantages with noncorrosive electrolytes and inexpensive simple salts (e.g., KCl and NaCl) as supporting electrolytes. In addition, neutral pH electrolytes suppress undesired side reactions for active species caused by protons and hydroxides at acidic and alkaline conditions[28,30]. So far methyl viologen (MV) aqueous RFBs have demonstrated the most stable cycling performances in neutral media[19,28,29,31]. Typically MV is employed as anolyte and ferrocene or (2,2,6,6-tetramethylpiperidin-1-yl)oxyl (TEMPO) derivatives as catholyte to develop high-voltage and stable pH neutral aqueous RFBs[19,29,30].

Therefore, the search and development of new water-soluble active materials for the improvement of neutral pH battery storage systems is increasingly significant and will continue to growth in the future. Tryptanthrin (TRYP) and its derivatives are a surprising family of compounds with biological and pharmacological activities[32–38] with the addition of displaying interesting redox properties due to the electron-accepting ability of the TRYP structure[39,40]. TRYP can be synthetically obtained from indigo, one of the most stable organic dyes[41–44]. TRYP shows two reversible waves with cathodic and anodic peaks, indicating two one-electron transfers[35,39]. Similar redox properties have also been reported for some azulene and benzo-annulated tryptanthrin derivatives[45,46].

In the effort to obtain new active materials presenting long-term stability for AORFBs, a water-soluble tryptanthrin, tryptanthrin sulfonic acid (TRYP-SO$_3$H), was synthesized and further tested with a home-built RFB set-up (Supplementary Fig. SI1). Electrochemical measurements at several pH values, with determination of the kinetic parameters, diffusion coefficient, and electron transfer rate constant, for TRYP-SO$_3$H at neutral pH values, were obtained. Charge–discharge processes and cell performance at neutral pH of (i) aqueous organometallic active materials and (ii) of all-organic active materials for RFB, combining this water-soluble tryptanthrin as the negative electrolyte (anolyte) with (i) potassium ferrocyanide and (ii) 4,5-dihydroxy-1,3-benzenedisulfonic acid disodium salt monohydrate (BQDS) as the positive electrolytes (catholyte), were evaluated.

## Results and discussion

**Synthesis and characterization.** Aiming the preparation of a water-soluble tryptanthrin derivative, a similar synthetic approach to the one reported by Pereira et al.[47] was followed. The chlorosulfonation of TRYP through electrophilic aromatic substitution with neat chlorosulfonic acid at 60 °C during 48 h under vigorous stirring and nitrogen atmosphere gave rise, after isolation, to a dark green solid, Fig. 1. The proton nuclear magnetic resonance ($^1$H NMR) spectrum of the reaction crude (Supplementary Fig. SI2), although with some unreacted TRYP (confirmed by TLC analysis), shows the peak around 8.70 ppm characteristic of hydrogen at the ortho position in this aromatic moiety, confirming that the electrophilic aromatic substitution take place at the position 8 or 2 of TRYP. Hydrolysis of the chlorosulfonation products in water, yield the desired tryptanthrin sulfonic acid derivatives. As expected, knowing that 2- and 8-positions of TRYP are activated for electrophilic aromatic substitution, being the 8-position more reactive than the 2-position[48], two isomers were obtained. The $^1$H NMR spectrum showed seven major peaks corresponding to seven hydrogen atoms, indicating that mono-substitution of TRYP occurs (Supplementary Fig. SI3). From the doublets at 8.69 and 8.67 ppm it is possible to calculate the isomer ratio with 85% of TRYP-8SO$_3$H and 15% of TRYP-2SO$_3$H. These results were further confirmed by high-performance liquid chromatography with a diode-array detector (HPLC-DAD) analysis (Supplementary Fig. SI4). The chromatogram showed two products with very similar polarity with retention times of 29.5 and 31.9 min and the percentage of total chromatogram integration of 72% and 14%, respectively (Supplementary Table SI1). Three independent experiments carried out following the described experimental procedure afford

**Fig. 1 Schematic synthetic route.** Tryptanthrin sulfonyl chloride derivatives and tryptanthrin sulfonic acid derivatives.

**Table 1 Electrochemical data including the oxidation ($E_{pa}$) and reduction ($E_{pc}$) potentials for TRYP-SO₃H in different pH of the supporting electrolyte at $T = 293$ K.**

| pH | Oxidation potential | Reduction potential |
|---|---|---|
| 0 | $E_{pa} = 0.097$ V | $E_{pc} = 0.034$ V |
| 7 | $E_{pa} = -0.406$ V | $E_{pc} = -0.507$ V |
| 13 | $E_{pa}^1 = -0.225$ V | $E_{pc}^1 = -0.325$ V |
|  | $E_{pa}^2 = -0.621$ V | $E_{pc}^2 = -0.802$ V |
|  | $E_{pa}^3 = -0.738$ V | $E_{pc}^3 = -0.930$ V |

$E_{pa}^1$, $E_{pa}^2$, and $E_{pa}^2$ = oxidation potential for the first, second, and third peak in the voltammogram in Supplementary Fig. SI9.
$E_{pc}^1$, $E_{pc}^2$, and $E_{pc}^2$ = reduction potential for the first, second, and third peak in the voltammogram in Supplementary Fig. SI9.

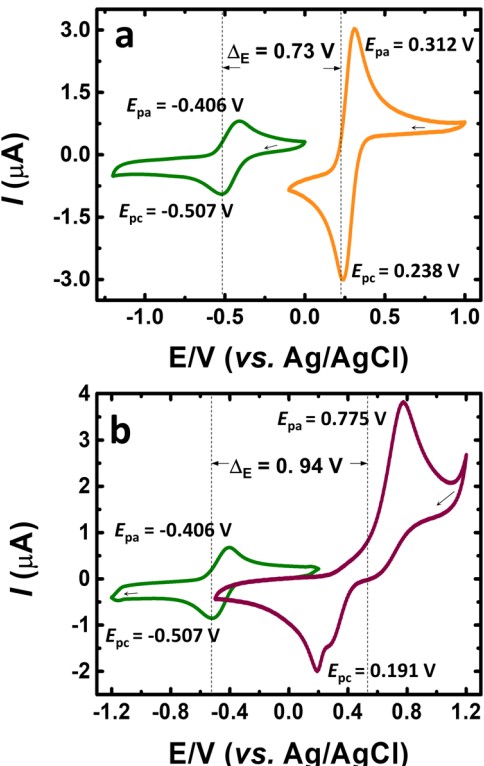

**Fig. 2 Cyclic voltammetry.** Cyclic voltammograms obtained with saturated N₂ at $v = 50$ mV s⁻¹ in 1.0 M KCl solution of electrolyte. **a** 1.0 mM TRYP-SO₃H (green trace) and 1.0 mM K₄[Fe(CN)₆]·3H₂O (orange trace). **b** 1.0 mM TRYSO₃H (green trace) and 1.0 mM BQDS (purple trace).

the final mixture of isomers in the same ratio (Supplementary Fig. SI5). The infrared (IR) spectrum shows bands at 775, 1200, 1220, and 1350 cm⁻¹ characteristic of hydrated sulfonic acid groups (Supplementary Fig. SI6). The compound was further characterized by high-resolution mass spectrometry (HRMS) and the obtained data were in agreement with the formula proposed m/z $[M - 1]^- = 327.0081$ calculated for $C_{15}H_7N_2O_5S$; found: 327.0078 (Supplementary Fig. SI7). The solvent-free methodology allows the synthesis of the water-soluble tryptanthrin-based active material in good yields and high purity.

**Tryptanthrin sulfonic acid—cyclic voltammetry.** Since there are only a few studies describing the redox properties of TRYP[39,40], the electrochemical behavior of this compound was initially performed by measuring cyclic voltammetry (CV) of TRYP in acetonitrile (MeCN) using 0.1 M of tetrabutylammonium hexafluorophosphate (NBu₄PF₆) as a supporting electrolyte. CV revealed two reversible peaks in the cathodic region and two reversible peaks in the anodic region, even after several scans, see Supplementary Fig. SI8. The reversible waves are separated by approximately 60 mV and these results are in agreement with previous studies[35,39].

To evaluate the electrochemical behavior of the water-soluble tryptanthrin sulfonic acid derivative (TRYP-SO₃H), CV measurements in an aqueous medium were obtained at different pH values. CVs are presented in Supplementary Fig. SI9 and the relevant electrochemical data, extracted from these, including the oxidation ($E_{pa}$) and reduction ($E_{pc}$) potentials, summarized in Table 1. The CVs obtained showed, at pH = 0, the occurrence of one reversible reduction peak at 0.034 V and one reversible oxidation peak at 0.097 V (Supplementary Fig. SI9a). At pH = 7, a similar behavior was also observed, with one cathodic peak at −0.507 V and one anodic peak at −0.406 V (Supplementary Fig. SI9b). The CV experiment at pH = 13 displays three peaks in the anodic region and three peaks in the cathodic region (Supplementary Fig. SI9c). From the obtained results it is visible that TRYP-SO₃H presents high reversible redox behavior at acidic and neutral pH, a mandatory condition for its use in aqueous RFBs.

In recent years, the development of neutral pH aqueous RFBs have stood out as promising RFBs technology for sustainable and safe energy storage[9,20,28,29]. One of the reasons is that neutral pH-based electrolytes are less-corrosive[19,29,49,50] and more eco-friendly when compared with the acidic or alkaline electrolytes that suffer from corrosion problems in cell stacks[4,13,51].

**Redox couples as active species in neutral pH aqueous redox flow battery cell tests.** To evaluate the viability of the redox couples consisting of TRYP-SO₃H/K₄[Fe(CN)₆] and TRYP-SO₃H/BQDS as the active species at neutral pH in the aqueous organometallic and all-organic active materials for RFB, the redox

reactivity was obtained from CVs (Fig. 2). Potassium ferrocyanide was chosen as catholyte due to the strong coordination of cyanide ions to the iron center, which makes the standard $[Fe(CN)_6]^{4-}$/$[Fe(CN)_6]^{3-}$ redox couple highly stable[52] and nontoxic[19]. In addition, this redox couple also showed ultra-stable cycling performance at neutral conditions, thus being more suitable for application in aqueous RFBs[53]. BQDS is an aromatic organic compound that belongs to the family of quinones and in recent years has been used as the positive active material in aqueous RFBs[26,54]. Due to a relatively high electrode potential (0.76 V)[26] and high solubility in sulfuric acid (0.65 M in 1.0 M H₂SO₄)[55], most of the reported studies with BQDS are in acidic medium. Our studies showed high solubility in KCl (1.28 M in 1.0 M KCl, see Supplementary Table SI2) and high electrode potential (0.94 V in KCl), demonstrating that BQDS can be viable as positive active material at neutral pH.

From Fig. 2a, it is possible to observe that the cell voltage of the redox reaction of K₄[Fe(CN)₆] and TRYP-SO₃H *vs.* Ag/AgCl is, at neutral pH, +0.27 V and −0.46 V, respectively, giving an expected cell potential of 0.73 V for the TRYP-SO₃H/K₄[Fe(CN)₆] redox couple. Compared with the well-known organic anolyte anthraquinone-2,7-disulfonic acid (2,7-AQDS), in neutral media, TRYP-SO₃H shows an identical cell potential: 0.76 V[25] *vs.* 0.73 V (this work). This demonstrates that TRYP-SO₃H can, when compared to quinone-type compounds, be considered viable as the active species for the anolyte. When the all-organic redox couple TRYP-SO₃H/BQDS (Fig. 2b) is used the cell voltage of BQDS *vs.* Ag/AgCl is 0.48 V, enhancing the positive shift in the redox potential and thus achieving a higher cell potential (0.94 V) when compared with K₄[Fe(CN)₆].

To further verify the correlation between the reaction rate of the catholyte/anolyte pair with the performance and stability of

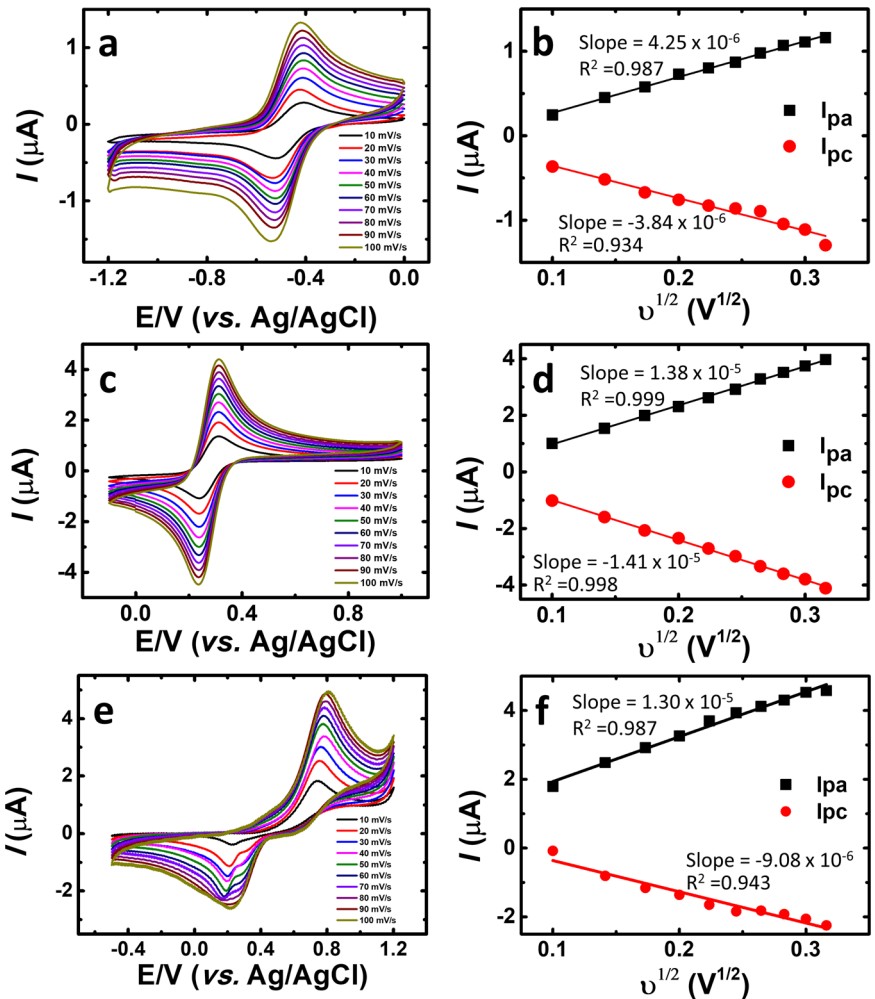

**Fig. 3 Determination of diffusion coefficient (D).** Cyclic voltammetry scan rate study obtained with saturated $N_2$ at various scan rates (10–100 mV s$^{-1}$) in 1.0 M KCl solution of electrolyte. **a** 1.0 mM TRYP-SO$_3$H at various scan rates. **b** plot of $i_p$a and $i_p$c over the square root of scan rates for 1.0 mM TRYP-SO$_3$H. **c** 1.0 mM K$_4$[Fe(CN)$_6$]·3H$_2$O at various scan rates. **d** plot of $i_p$a and $i_p$c over the square root of scan rates for 1.0 mM K$_4$[Fe(CN)$_6$]·3H$_2$O. **e** 1.0 mM BQDS at various scan rates. **f** plot of $i_p$a and $i_p$c, over the square root of scan rates for 1.0 mM BQDS. In **b**, **d**, and **f** black squares and line for oxidative reaction and red squares and line for reductive reaction.

the RFBs, the kinetic parameters, diffusion coefficient ($D$), and electron transfer rate constant ($k_o$), were quantified. The Randles–Sevcik equation[56] (Eq. 1) was used to calculate the diffusion coefficient ($D$ in cm$^2$ s$^{-1}$)

$$i_p = 0.4463 nFAC^0 \left(\frac{nFD}{RT}\right)^{1/2} \qquad (1)$$

where, $i_p$ is the cathodic or anodic peak current, $n$ is the number of electrons, $F$ the Faraday's constant (C mol$^{-1}$), $A$ is the electrode surface area (in cm$^2$), $C^0$ is the concentration of the oxidative or reductive species (mol cm$^{-3}$), $R$ the ideal gas constant (J K$^{-1}$ mol$^{-1}$), $T$ the temperature (K) and $v$ is scan rate in V s$^{-1}$. To measure the diffusion coefficients of the K$_4$[Fe(CN)$_6$]/TRYP-SO$_3$H and BQDS/TRYP-SO$_3$H redox couples, CV experiments together with the plot of the peak current $i_p$ as a function of the square root of the scan rate $v^{1/2}$ (Fig. 3) are needed to obtain the Randles–Sevcik equation parameters. A linear fit ($i_p =$ slope $\times v^{1/2}$) yields the slope of the cathodic and anodic peaks from which the diffusion coefficient can be obtained (experimental details in Supplementary Tables SI3–SI5).

The electron transfer rate constant ($k_o$) of K$_4$[Fe(CN)$_6$]/TRYP-SO$_3$ and BQDS/TRYP-SO$_3$ redox couples were also estimated by using the Nicholson method using the $D$ values previously

obtained[57]. The relationship between the $k_o$ and the Nicholson dimensionless number ($\Psi$) is given by Eq. 2

$$k_o = \left(\pi D \left(\frac{nF}{RT}\right)v\right)^{\frac{1}{2}} \times \Psi \qquad (2)$$

Using BQDS (1.0 mM), the $D$ and $k_o$ values ($5.38 \times 10^{-7}$ and $2.71 \times 10^{-4}$ cm s$^{-1}$) were close to those obtained for TRYP-SO$_3$H ($6.98 \times 10^{-8}$ and $3.62 \times 10^{-4}$ cm s$^{-1}$) when compared to the rate constants values for K$_4$[Fe(CN)$_6$] ($6.63 \times 10^{-6}$ and $1.24 \times 10^{-2}$ cm s$^{-1}$), see experimental details in Supplementary Tables SI6–SI8. The electron transfer rate constant for TRYP-SO$_3$H is in the range found for quinones and anthraquinones used in aqueous RFBs[9]. This shows that despite some differences in the diffusion coefficients and of the rate constants of the redox couples, both redox pairs are suitable as active materials for aqueous organometallic and all-organic RFBs. Based on the above results, both organometallic and all-organic redox couples were tested in a cell stack.

**Neutral pH aqueous redox flow battery cell tests.** Two single cells were assembled to evaluate the properties of the TRYP-SO$_3$H/K$_4$[Fe(CN)]$_6$ and TRYP-SO$_3$H/BQDS redox pairs for aqueous organometallic and all-organic active materials for RFB

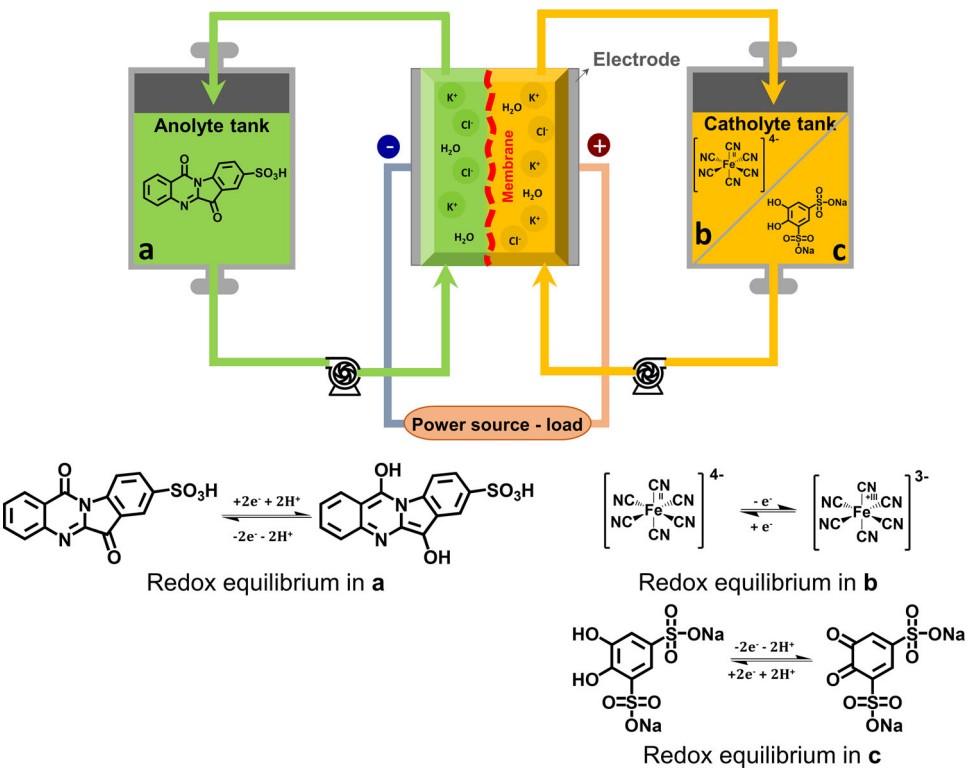

**Fig. 4 Schematic illustration of RFBs cell tests.** Aqueous organometallic and all-organic RFB cell using TRYP-SO$_3$H and K$_4$[Fe(CN)$_6$]/BQDS redox couples in neutral pH, and redox reaction mechanisms of TRYP-SO$_3$H (**a**), K$_4$[Fe(CN)$_6$] (**b**), and BQDS (**c**).

working at neutral pH (Fig. 4). Solubility measurements showed that TRYP-SO$_3$H has a solubility of approximately 0.12 M in 1.0 M KCl at 293 K (see Supplementary Table SI2). Therefore, the concentration of active materials used is relatively low to avoid the possible precipitation of active materials during cycling. Representative galvanostatic charge–discharge curves for fifty complete cycles of the flow cells obtained for the TRYP-SO$_3$H/ K$_4$[Fe(CN)$_6$] and TRYP-SO$_3$H/BQDS couples with a current density applied of 5.2 and 2.6 mA cm$^{-2}$, respectively, are shown in Fig. 5. The current densities were limited to low values due to resistance of the electrolyte to avoid excess ohmic loss. The cells can be charged and discharged within the selected potential window (the cut-off potential set between 0.2 and 1.2 V for TRYP-SO$_3$H/K$_4$[Fe(CN)$_6$] and between 0.5 and 1.5 V for TRYP-SO$_3$H/BQDS) with reproducible cycles. The charge–discharge profiles for the two RFB are slightly different. Indeed, whereas with the TRYP-SO$_3$H/K$_4$[Fe(CN)$_6$] redox couple the interval period between charging and discharging is longer for the first cycle than it is for the others (Fig. 5a), with the TRYP-SO$_3$H/ BQDS redox couple all charge–discharge cycle are identical (Fig. 5b) and stable from the first cycle till the end of the cell test of ~29 h (50 cycles). It is also noticeable that the charging and discharging time using the all-organic active materials is four times longer than with the organometallic active materials (see Supplementary Fig. SI10).

The obtained average discharge energy density and average discharge capacity of the organometallic active materials cell were 0.014 Wh L$^{-1}$ and 1.17 mAh, respectively, while that of the all-organic TRYP-SO$_3$H/BQDS redox couple is highest, with values of 0.046 Wh L$^{-1}$ and 2.65 mAh. Long-time capacity stability is a vital characteristic for aqueous RFBs. Indeed, while for the organometallic active materials cell (Fig. 6a) during fifty complete cycles (~7 h) there is a decrease in the charge and discharge energy density and capacity, with capacity retention falling to

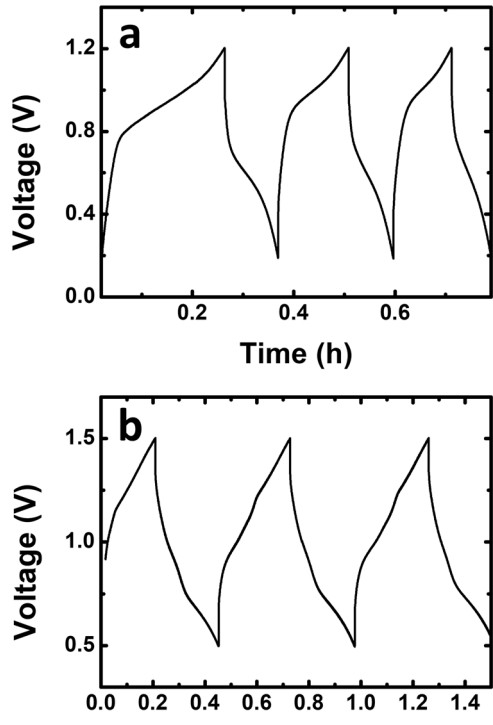

**Fig. 5 Charge–discharge cycling performance.** Representative galvanostatic charge–discharge curves of neutral pH aqueous organometallic and all-organic active materials for RFB single cells using 5.0 mM of TRYP-SO$_3$H in 1.0 M KCl solution as supporting electrolyte measured at the third cycle. **a** 10.0 mM of K$_4$[Fe(CN)$_6$]·3H$_2$O as catholyte. **b** 5.0 mM of BQDS as catholyte.

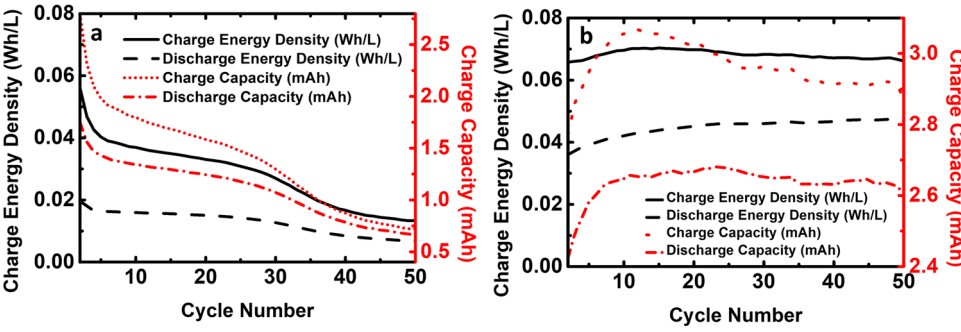

**Fig. 6 Cell performance.** Charge–discharge energy density and charge–discharge capacity plots of neutral pH aqueous organometallic and all-organic active materials for RFB single cells using 5.0 mM of TRYP-SO$_3$H in 1.0 M KCl as supporting electrolyte. **a** 10.0 mM of K$_4$[Fe(CN)$_6$]·3H$_2$O as catholyte. **b** 5.0 mM of BQDS as catholyte.

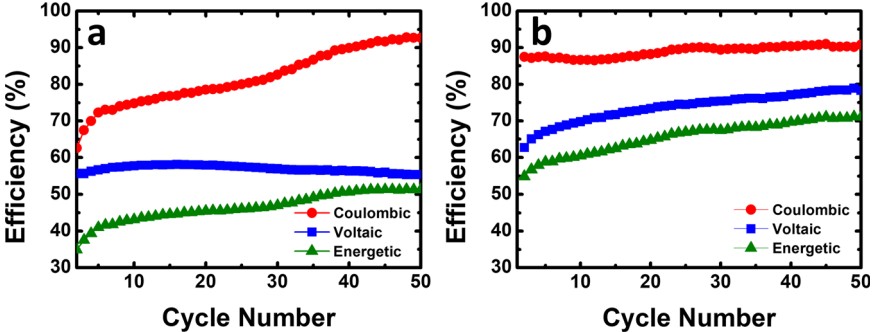

**Fig. 7 System efficiencies.** Coulombic efficiency, voltaic efficiency, and energetic efficiency plots of neutral pH aqueous organometallic and all-organic active materials for RFB single cells using 5.0 mM of TRYP-SO$_3$H in 1.0 M KCl as supporting electrolyte. **a** 10.0 mM of K$_4$[Fe(CN)$_6$]·3H$_2$O as catholyte. **b** 5.0 mM of BQDS as catholyte.

38% of its original value (6.309–2.409 C) over fifty cycles (see Supplementary Fig. SI11a). In general, non-capacity-related coulombic efficiency loss (capacity fades over the cycles but coulombic efficiency increases, Supplementary Fig. SI11a) mainly arises from electrolyte side reactions[58]. In the all-organic active materials cell (Fig. 6b) the values steadily increase until ~10 cycles and further stabilizes to a total number of fifty cycles (~29 h). Stable capacity retention was observed with more than 98% total capacity retention after forty cycles (Supplementary Fig. SI11b).

The coulombic, voltaic, and energetic efficiencies of the aqueous organometallic and all-organic active materials *vs.* the number of cycles in the flow cell are shown in Fig. 7. Coulombic and voltage efficiencies are the properties that better evaluate the performance of RFBs. For the neutral pH aqueous organometallic active materials RFB (Fig. 7a) the values found for coulombic, voltaic, and energetic efficiencies were 80%, 57%, and 46%, respectively. In the all-organic active materials cell, the obtained values were higher, displaying an interesting cycling performance with 89% coulombic efficiency, 75% of voltaic efficiency, and 67% of energy efficiency. It can be seen that at neutral pH the aqueous all-organic active materials flow cell presents notable cycling retention (98%) and coulombic efficiencies above ~90% through several cycles (see Supplementary Fig. SI11b).

In order to quantify the electrochemical stability of the active materials and to evaluate the possibility of crossover or chemical degradation of the aqueous organometallic and all-organic active materials for RFB during cycling, electrochemical impedance spectra were measured before and after charge–discharge cycles, see Figs. 8 and 9. From the voltaic efficiency, it is possible to assess losses through electrolyte crossover[59]. During the cell test, there were no vestiges of crossover through the membrane. After the cell measurement, the CV measurement performed to the catholyte

(K$_4$[Fe(CN)$_6$]) (Fig. 8a) does not differ from the initial one (Fig. 2) and no peaks in the CV of the K$_4$[Fe(CN)$_6$] electrolyte in the range of the CV of TRYP-SO$_3$H (see Supplementary Fig. SI12) could be observed. According to Fig. 8b, and after full cell tests, a change in the voltammogram profile of TRYP-SO$_3$H is seen, with the peak maxima shifting to more negative potentials together with new anodic and cathodic peaks. These results may help to explain the changes observed in the values of the charge–discharge cycles with time, as well as the observed battery capacity loss, which is likely related to some chemical modification of TRYP-SO$_3$H after 50 cycles. From Fig. 9a, it is visible that the cyclic voltammetry curves of the BQDS electrolyte, obtained before and after the charge–discharge studies, are also different. After finalizing the experiment with the RFB, the BQDS showed an additional redox peak. The instability of BQDS has already been reported in an acidic medium, where this compound can undergo a Michael addition reaction with water leading to trihydroxybenzene derivatives[17,54,55,60]. This reaction may also explain the obtained results at neutral pH values which is further reflected in the capacity decrease. According to Fig. 9b, before cycling, the TRYP-SO$_3$H electrolyte has a current intensity of the reduction and oxidation peak much higher than after fifty charge–discharge cycles indicating loss of reversibility.

Crossover through the membrane was also investigated. After the charge–discharge experiment ended, no peaks in the cyclic voltammogram of the BQDS electrolyte, in the range of the TRYP-SO$_3$H peaks, could be observed (see Supplementary Fig. SI13), indicating the absence of crossover of the TRYP-SO$_3$H electrolyte. Another possibility for the capacity fading is the leakage from the cell stack system. After cell cycling, both cells were disassembled and no coloration was found on the gaskets, indicating the absence of electrolyte leakage.

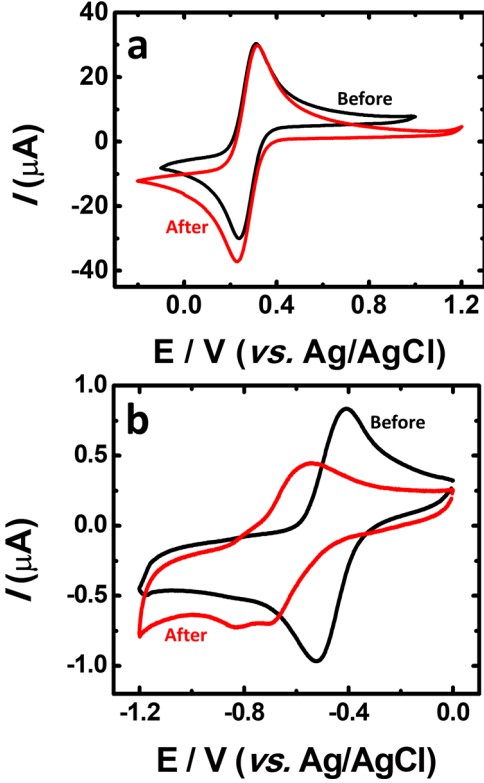

**Fig. 8 Cyclic voltammetry curves before and after aqueous organometallic RFB cell test.** Comparison between the catholyte ($K_4[Fe(CN)_6] \cdot 3H_2O$) (**a**) and anolyte (TRYP-$SO_3H$) (**b**) solutions before (black curve) and after (red curve) aqueous organometallic active materials for RFB full cell test using 1.0 M KCl as supporting electrolyte. Solutions with saturated $N_2$, $v = 50$ mV s$^{-1}$.

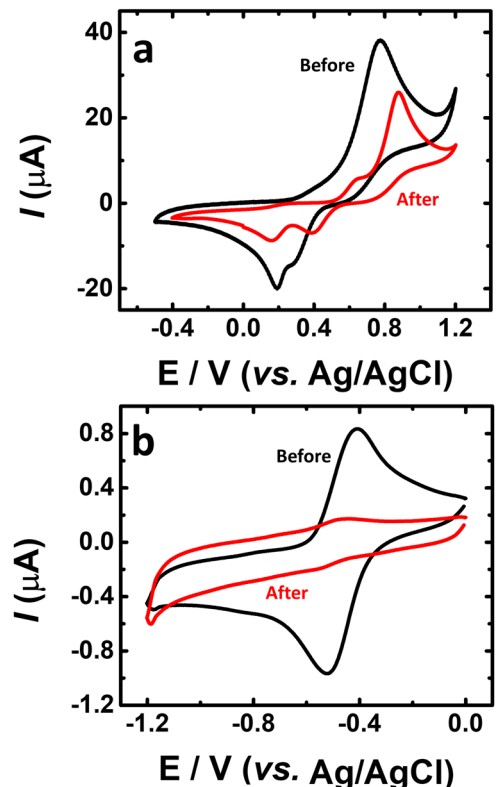

**Fig. 9 Cyclic voltammetry curves before and after aqueous all-organic RFB cell test.** Comparison between the catholyte (BQDS) (**a**) and anolyte (TRYP-$SO_3H$) (**b**) solutions before (black curve) and after (red curve) aqueous organometallic active materials for RFB full cell test using 1.0 M KCl as supporting electrolyte. Solutions with saturated $N_2$, $v = 50$ mV s$^{-1}$.

According to the results, the all-organic TRYP-$SO_3H$/BQDS redox couple showed better battery performance than the organometallic TRYP-$SO_3H$/$K_4[Fe(CN)_6]$ redox couple, confirming the role of BQDS in the promotion of the electrochemical performance of the cell.

In order to study the effect of the concentration of the all-organic active materials in the stability and performance of the RFB, the concentration of the redox pair was increased to 0.1 M, a concentration close to the solubility limit of TRYP-$SO_3H$ (0.12 M in 1.0 M KCl, see Supplementary Table SI2). The data is presented in Figs. 10 and 11 and can be summarized as follows: (i) with the augment on the concentration of TRYP-$SO_3H$ and BQDS, the active materials remain well dissolved (see Supplementary Table SI2) and operated well during 50 cycles (Fig. 10); (ii) as the concentration of TRYP-$SO_3H$ and BQDS increased, the coulombic efficiency reaches 95% (Fig. 11); (iii) with 0.1 M of active materials, the discharge energy density and capacity values are, however, lower (Fig. 12) when compared with the highest values displayed by other systems working at neutral pH values[19,20,28,50,61]. The rise of the ohmic resistance and viscosity of the redox couple with the concentration may explain the results obtained.

## Conclusions

In the search for new water-soluble active materials for aqueous RFBs, the use of a water-soluble tryptanthrin (TRYP-$SO_3H$), was found to be a promising approach for sustainable development for these applications. CV measurements in an aqueous medium showed that TRYP-$SO_3H$ exhibits high redox reversibility with particular emphasis at neutral pH values. As a result, two cells,

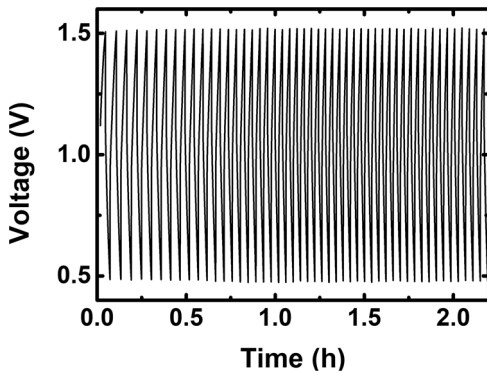

**Fig. 10 Charge–discharge curves vs. time.** Galvanostatic charge–discharge curves of neutral pH aqueous all-organic active materials for RFB single cell using 0.1 M of TRYP-$SO_3H$ as anolyte and 0.1 M of BQDS as catholyte in 1.0 M KCl solution as supporting electrolyte measured during 50 cycles.

organometallic and all-organic, were assembled by using TRYP-$SO_3H$ as anolyte and $K_4[Fe(CN)_6]$ and BQDS as catholyte. At neutral pH values both organometallic and all-organic RFBs display promising results; yet, the all-organic TRYP-$SO_3H$/BQDS RFB demonstrates significant improvement in the performance and merit of the cell, with higher average coulombic (89%), voltaic (75%), and energy (67%) efficiencies and stable charge–discharge curves during ~29 h. The present study highlights a new perspective on screening water-soluble tryptanthrin derivatives as organic active materials and underlines the possibility to develop

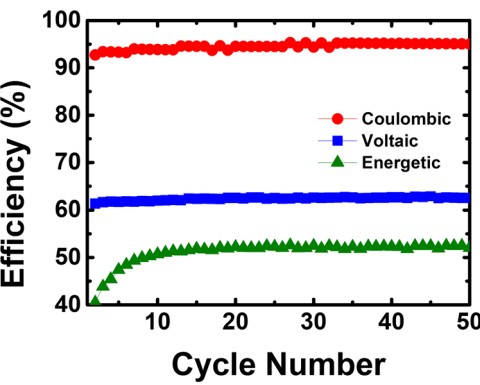

**Fig. 11 Cycling efficiencies vs. cycle number.** Coulombic efficiency, voltaic efficiency, and energetic efficiency plots of neutral pH aqueous all-organic active materials for RFB single cell using 0.1 M of TRYP-SO$_3$H as anolyte and 0.1 M of BQDS as catholyte in 1.0 M KCl as supporting electrolyte.

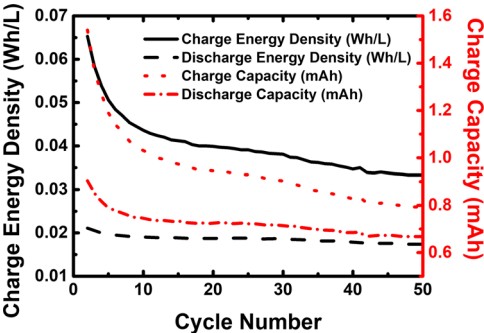

**Fig. 12 Energy density and capacity vs cycle number.** Charge-discharge energy density and charge–discharge capacity plots of neutral pH aqueous all-organic active materials for RFB single cell using 0.1 M of TRYP-SO$_3$H and 0.1 M of BQDS as catholyte in 1.0 M KCl as supporting electrolyte.

low-cost and environmentally friendly AORFBs (working at neutral pH) using sustainable redox-active organic molecules.

## Methods

**Chemicals.** Tryptanthrin (TRYP) was synthesized following the methodology recently reported by our group[41,62]. Chlorosulfonic acid (ClSO$_3$H) was purchased from Merck, sodium bicarbonate (NaHCO$_3$), potassium ferricyanide (K$_3$[Fe (CN)$_6$]) and sodium hydroxide (NaOH) were purchased from José Manuel Gomes dos Santos Lda, potassium ferrocyanide (K$_4$[Fe(CN)$_6$].3H$_2$O) was purchased from Baker Analyzed Reagent, tetrabutylammonium hexafluorophosphate and 4,5-dihydroxy-1,3-benzenedisulfonic acid disodium salt monohydrate (BQDS) were purchased from Sigma-Aldrich, potassium chloride (KCl) was purchased from Panreac, hydrogen peroxide (H$_2$O$_2$) was purchased from Valente e Ribeiro Lda and sulfuric acid (H$_2$SO$_4$) from Labkrem. All the reagents were used without further purification. The solvents, pro analysis (P.A.) quality, were used as purchased. Ultra-pure water at pH = 5.4, was purified using Direct Q3 Merck Millipor equipment.

**Characterization techniques.** Nuclear magnetic resonance (NMR) spectra were recorded at room temperature in deuterated chloroform (CDCl$_3$) solutions on a Bruker Avance III spectrometer operating at 400.13 MHz for $^1$H. Tetramethylsilane was used as an internal standard. High-resolution mass spectrometry (HRMS) was performed on a Bruker microTOF-Focus mass spectrometer equipped with an electrospray ionization time-of-flight (ESI–TOF) source. Tryptanthrin sulfonic acid was analyzed in an analytical Elite Lachrom high-performance liquid chromatography system with L-2455 diode array detector (HPLC-DAD), L-23000 column oven (stationary phase Purospher® STAR RP-18 endcapped (5 µm) column from Merck), L-2130 Pump and an L-2200 autosampler. IR spectroscopy was recorded using the Thermo Fisher Nicolet 6700 FT*IR* spectrometer.

**Synthetic procedure.** In the synthesis, 300 mg of TRYP (1.2 mmol) was placed in a bottom flask in a paraffin bath at 60 °C. Keeping an inert atmosphere, chlorosulfonic acid (3 mL, 45 mmol) was added and the mixture was stirred for 48 h.

The solution was left to cool in ice and neutralized with a saturated solution of sodium bicarbonate (NaHCO$_3$). The garnet solid obtained was filtrated and washed with water and the solid was dried at 45 °C overnight to yield 173 mg of a dark green solid. NMR analysis of the reaction crude showed that the dark green solid consisted in a mixture of, tryptanthrin sulfonyl chloride and some unreacted TRYP. The dark green mixture (100 mg) was suspended in 50 mL of water and further heated at reflux for 48 h until a green solution was obtained. After cooling to room temperature, the solution was filtrated to remove a trace of non-dissolved TRYP and the solvent was evaporated under vacuum. The green solid obtained was dried at 45 °C for 24 h to yield 73 mg of a mixture of two isomers, 6,12-dioxo-6,12-dihydroindolo[2,1-b]quinazoline-8-sulfonic acid, tryptanthrin 8-sulfonic acid (TRYP-8SO$_3$H), and 6,12-dioxo-6,12-dihydroindolo[2,1-b]quinazoline-2-sulfonic acid, tryptanthrin 2-sulfonic acid (TRYP-2SO$_3$H) in a 85:15 ratio. $^1$H NMR (CDCl$_3$, 400 MHz,), δ 8.69 (d, $J$ = 1.6 Hz, 1H), 8.44 (ddd, $J$ = 8.0 Hz, $J$ = 1.6 Hz, $J$ = 0.4 Hz, 1H), 8.05 (ddd, $J$ = 8.4 Hz, $J$ = 1.2 Hz, $J$ = 0.8 Hz, 1H), 7.87 (m, 2H), 7.70 (dt, $J$ = 8.4 Hz, $J$ = 1.2 Hz, 1H), 7.40 (dd, $J$ = 8.0 Hz, $J$ = 1.6 Hz, 1H) ppm. HRMS (ESI–TOF-MS): m/z [M − 1]$^-$ = 327.0081 calculated for C$_{15}$H$_7$N$_2$O$_5$S; found: 327.0078. IR (KBr pellets) wavenumber (cm$^{-1}$): 775, 1200, 1220, 1350, 1430, 1590, 1680, and 1730 cm$^{-1}$. HPLC-DAD: Stationary phase Purospher® STAR RP-18 endcapped (5 µm). Eluent: potassium acetate (pH = 3)(A) and MeCN:H$_2$O (50:50)(B) (from 100:0 to 30:70(A/B v/v)); 5% increment of B each min for 12 min; 5% increment of B each 2 min until 22 min and 30:70 (A/B v/v) ratio for 12 min. The flow rate of 0.8 mL/min in the first 9 min and then 0.4 mL/min until 35 min. L-2455 Diode Array Detector (400 nm). Percentage of total chromatogram integration at retention time 29.49 min and 31.89 min of 72% and 14%, respectively.

**Solubility measurements.** The solubility of the active materials at room temperature was measured in 1.0 M KCl. To ascertain the solubility limit a defined amount of each compound was first weighed and transferred to a glass tube after which an increasing volume of 1.0 M KCl solution was added in intervals of 10–100 µL. After each addition, the test tube was ultrasonic for at least 10 min at room temperature. Then the tube was inspected visually for an undissolved sample and the addition of 1.0 M KCl solution continued until the sample seemed just dissolved. Dissolved samples were left to stand for ~12 h and visually inspected. The volumes just before and after the full dissolution was used to calculate the solubility interval.

**Cyclic voltammetry.** CV experiments were carried out using an Autolab potentiostat/galvanostat PGSTAT204 running with NOVA 2.1 software and a three-electrode system in a one-compartment electrochemical cell with a capacity of 10 mL. A glassy carbon electrode (GCE) ($d$ = 3 mm) was the working electrode, GC ($d$ = 1.6 mm) wire the counter electrode, and Ag/Ag+ (0.01 M silver nitrate (AgNO$_3$) in 0.1 M tetrabutylammonium hexafluorophosphate (NBu$_4$PF$_6$) in MeCN) the reference electrode for nonaqueous solutions. In an aqueous medium Ag/AgCl (3.0 M KCl) was the reference electrode. The GCE was polished with appropriate polishing pads using first aluminum oxide particle size 0.3 µm and then aluminum oxide particle size 0.075 µm (polish in a Fig. 8 motion) before each electrochemical experiment. After polishing, the electrode was rinsed thoroughly with Milli-Q water and the electrode was sonicate in a container with Milli-Q water and ethanol (50:50 v/v) for 5 min. Following this mechanical treatment, the GCE was placed in buffer supporting electrolyte, and differential pulse voltammograms were recorded until a steady-state baseline voltammogram was obtained. This procedure ensured very reproducible experimental results.

TRYP sample was dissolved in 10 mL of acetonitrile (MeCN) with 0.1 M of NBu$_4$PF$_6$ as the supporting electrolyte. The compound was prepared at a concentration of 1.0 mM and was degassed with nitrogen for 10 min prior to analysis. A solution of 1.0 mM of ferrocene with 0.1 M of NBu$_4$PF$_6$ in 10 mL of MeCN was used as standard and also degassed with nitrogen for 10 min prior to analysis. An atmosphere of nitrogen was maintained during the voltammetric experiment and the sample was run at a scan rate of 50 mV s$^{-1}$.

To evaluate open-circuit voltage (OCV) between TRYP-SO$_3$H and K$_3$[Fe(CN)$_6$] at different pH values, the solutions were prepared at a concentration of 2.0 mM and degassed with nitrogen for 10 min prior to analysis. Depending on the pH experiments different electrolytes were used, for pH = 0 (1.0 M H$_2$SO$_4$) solution, for pH = 7 (1.0 M KCl) solution, and pH = 13 (1.0 M NaOH) solution. Potassium ferricyanide (K$_3$[Fe(CN)$_6$]·3H$_2$O) (1.0 mM solution) with 1.0 M of the proper electrolyte in 10 mL of Milli-Q water was used as standard and also degassed with nitrogen for 10 min prior to analysis. Lastly, to measure the OCV for BQDS, 1.0 mM was dissolved into 1.0 M KCl solution. An atmosphere of nitrogen was maintained during the voltammetric experiments and the samples were run at a scan rate of 50 mV s$^{-1}$.

For the determination of the kinetic parameters, such as diffusion coefficient ($D$) and electron transfer rate constant ($k_o$), a solution of TRYP-SO$_3$H (1.0 mM) was dissolved in 10 mL of Milli-Q water with 1.0 M KCl as the supporting electrolyte. First, a solution of 1.0 mM of K$_4$[Fe(CN)$_6$] with 1.0 M of KCl in 10 mL of Milli-Q water was used as catholyte. Second, the catholyte was prepared by dissolving 1.0 mM of BQDS into 10 mL of 1.0 M KCl solution. All the solutions were degassed with nitrogen for 10 min prior to analysis. An atmosphere of nitrogen was maintained during the voltammetric experiments and the samples

were run at a scan rate of 10–100 mV s$^{-1}$ range. All the CV experiments were carried out at $T = 293$ K.

**Cell test measurements**. The flow cell for the aqueous RFBs was assembled with two steel end frame plates and two copper current collectors, held in place using two carbon electrolyte chambers. Graphite foil was used to form a flexible interconnect to the copper end-plate. Ethylene propylene diene monomer rubber gaskets were positioned on top of the carbon plate and the carbon felt electrodes (Alfa Aesar, 3.18 mm) were positioned within the gaskets. A piece of Nafion® perfluorinated membrane (Aldrich, nafion® 115) was sandwiched between carbon felts and the battery was compressed using tie-bolts. Each carbon chamber was connected with an electrolyte reservoir using a piece of Viton-type tube. The electrolyte reservoirs were 100 mL glass containers. The active area of the cell was 4 cm$^2$. A Master® L/S® peristaltic pump (Cole-Parmer, Easy-load II, Model 77202-60) was used to press sections of Masterflex tubing to circulate the electrolytes through the electrodes at a flow rate of 30 mL min$^{-1}$. Both reservoirs were purged with nitrogen to remove O$_2$ for 30 min and an atmosphere of nitrogen was maintained during the cell cycling. The flow cell was galvanostatically charged/discharged at room temperature and measurements were carried out with a current density applied of 20 mA using 0.2 and 1.2 V cut-off potentials in the first test (TRYP-SO$_3$H/K$_4$[Fe (CN)$_6$]) and a current density applied of 10 mA using 0.5 and 1.5 V cut-off potentials in the second test (TRYP-SO$_3$H/BQDS). The charge-discharge curves were recorded using an Autolab potentiostat/galvanostat PGSTAT204 running with NOVA 2.1 software. In the flow cell experiments, it was ensured that the same number of equivalents of the catholyte and anolyte was used on both sides. The negative electrolyte was prepared by dissolving TRYP-SO$_3$H (5.0 mM) in 50 mL of Milli-Q water with 1.0 M KCl as the supporting electrolyte. First, a solution of 10.0 mM of K$_4$[Fe(CN)$_6$] with 1.0 M of KCl in 50 mL of Milli-Q water was used as a positive electrolyte. Second, the catholyte was prepared by dissolving 5.0 mM of BQDS into 50 mL of 1.0 M KCl solution. Finally, the negative electrolyte was prepared by dissolving TRYP-SO$_3$H (0.1 M) in 50 mL of Milli-Q water with 1.0 M KCl as the supporting electrolyte and the catholyte was prepared by dissolving 0.1 M of BQDS into 50 mL of 1.0 M KCl solution. All the solutions were degassed with nitrogen for 30 min prior to analysis and an atmosphere of nitrogen was maintained during cell cycling. All experiments were carried out at room temperature ($T = 293$ K).

**Membrane treatment**. The Nafion perfluorinated membrane was initially emerged in Milli-Q water at 80 °C for 15 min and then put into 5% hydrogen peroxide solution (H$_2$O$_2$) for 30 min to remove organic impurities. In the next step in order to cleanse the metallic impurities, the membrane was put into 0.05 M KCl solution for one hour (after 30 min the KCl solution was changed). In the last step, the membrane was put into Milli-Q water for one hour, changing the water every 15 min. After pre-treatment, the membrane was placed in Milli-Q water to avoid further contaminations.

**Carbon felt thermal treatment**. A piece of carbon felt was heated at 400 °C for 24 h in a muffle furnace Vulcan 3-550. Then, the temperature of the muffle furnace was lowered to room temperature and the carbon felt was removed and properly stored until further use.

## Data availability

Supplementary Figures SI1–SI13 and Tables SI1–SI8 are provided as a Source Data File. Other relevant data are available from the corresponding author upon reasonable request.

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

## Acknowledgements
This work was supported by Projects "Hylight" (No. 031625) 02/SAICT/2017, PTDC/QUI-QFI/31625/2017, which is funded by the Portuguese Science Foundation (FCT) and Compete Centro 2020 and project "SunStorage - Harvesting and storage of solar energy", reference POCI-01-0145-FEDER-016387, funded by European Regional Development Fund (ERDF), through COMPETE 2020—Operational Program for Competitiveness and Internationalization (OPCI), and by national funds, through FCT. The Coimbra Chemistry Center is supported by FCT, through Projects UIDB/00313/2020, and UIDP/00313/2020. FCT is also gratefully acknowledged for a Ph.D. grant to D.P. (ref SFRH/BD/74351/2010). D.P. also acknowledges project "SunStorage—Harvesting and storage of solar energy" for a research grant. We also acknowledge the UC-NMR facility for obtaining the NMR data (www.nmrccc.uc.pt).

## Author contributions
J.S.S.M. formulated the project. D.P. synthesized, with the help of M.P., in the synthetic strategy, the compound and collected the HPLC-DAD data. D.P. and M.P. analyzed the NMR, HPLC-DAD, and Infrared data. D.P. measured the solubility, collected, and analyzed the electrochemical data, and performed the RFBs cell tests. J.S.S.M. wrote the article with the contribution of D.P. and M.P. All authors contributed to the overall scientific interpretation.

## Competing interests
A patent request deposit no. 20201000045444 (IPN) was made on 11 September 2020 with the contribution of all authors.
