## [Peer Review File · Communications Chemistry]

This manuscript has been previously reviewed at another Nature Research journal. This document only contains reviewer comments and rebuttal letters for versions considered at Communications Chemistry.

Reviewers' comments:

Reviewer #1 (Remarks to the Author):

The authors have fully addressed my comments and improved the manuscript a lot, I recommend it for publication.

Reviewer #2 (Remarks to the Author):

Issues raised have been well-addressed in the revised version. I recommend for publication in Communications Chemistry.

Reviewer #3 (Remarks to the Author):

Comments:

This manuscript presents the synthesis and flow battery studies of tryptanthrin sulfonic acid (TRYP-SO₃H). These authors provided details for characterization and electrochemical studies of the compound, which is useful for future reference. Flow battery studies using ferrocyanide and BQDS catholytes were reported. The battery data are not very exciting compared to status of the AORFB field. However, the effort to develop new redox active compounds for energy storage is appreciated. This manuscript can be considered after addressing the comments below.

1. As the focus of this study is on pH neutral AORFBs, I would like the authors to provide a more informative summary on the status of pH neutral AORFBs, particularly viologen AORFBs which represent the state of the art AORFBs. I agree with the team that it is more attractive to develop pH neutral AORFBs compared to acidic and alkaline AORFBs which are subject to chemical degradation under corrosive conditions. In this regard, the author should highlight/compare the performance of pH neutral, acidic, and alkaline AORFBs in the introduction. I would recommend the authors to read and cite a very valuable review article on organic flow batteries (ACS Energy Lett. 2019, 4, 2220-2240.).
2. I don't find the solubility of the tryptanthrin in the text. This data should be provided and discussed.
4. In Figure 5, capacity data should be provided and discussed in the main text.
5. It seems the tryptanthrin AORFBs were tested at low concentrations. Is it possible to try a higher concentration such as 0.5 M? If not, then I suggest to the authors provide NMR and/or other spectroscopic studies to gain possible degradation of the compound.

Point-by-point response to the referees' comments

Reviewer #1 (Remarks to the Author):

The authors have fully addressed my comments and improved the manuscript a lot, I recommend it for publication.

Reply from the authors: We much appreciate the comments of this referee.

Reviewer #2 (Remarks to the Author):

Issues raised have been well-addressed in the revised version. I recommend for publication in Communications Chemistry.

Reply from the authors: We much appreciate the comment of this referee

Reviewer #3 (Remarks to the Author):

Comments:

This manuscript presents the synthesis and flow battery studies of tryptanthrin sulfonic acid (TRYP-SO₃H). These authors provided details for characterization and electrochemical studies of the compound, which is useful for future reference. Flow battery studies using ferrocyanide and BQDS catholytes were reported. The battery data are not very exciting compared to status of the AORFB field. However, the effort to develop new redox active compounds for energy storage is appreciated. This manuscript can be considered after addressing the comments below.

1. As the focus of this study is on pH neutral AORFBs, I would like the authors to provide a more informative summary on the status of pH neutral AORFBs, particularly viologen AORFBs which represent the state of the art AORFBs. I agree with the team that it is more attractive to develop pH neutral AORFBs compared to acidic and alkaline AORFBs which are subject to chemical degradation under

corrosive conditions. In this regard, the author should highlight/compare the performance of pH neutral, acidic, and alkaline AORFBs in the introduction. I would recommend the authors to read and cite a very valuable review article on organic flow batteries (ACS Energy Lett. 2019, 4, 2220-2240.).

Reply from the authors: We thank the reviewer for his/her comments. Following the reviewer suggestions the introduction was carefully revised and the reference indicated incorporated as well as others that were used from that revision work. In particular, in the revised version of the manuscript the following text was added:

“To overcome the limitation of the corrosive electrolytes (acidic or alkaline media), development of neutral aqueous RFBs has emerged over the years.^{9,20,25,28,29} Neutral AORFBs are more ecofriendly and have outstanding advantages with non-corrosive electrolytes and inexpensive simple salts (e.g. KCl, NaCl) as supporting electrolytes. In addition, neutral pH electrolytes suppress undesired side reactions for active species caused by protons and hydroxides at acidic and alkaline conditions.^{28,30} So far methyl viologen (MV) aqueous RFBs have demonstrated the most stable cycling performances with capacity retention up to 99.99% in neutral media.^{19,28,29,31} Typically MV is employed as anolyte and ferrocene or (2,2,6,6-tetramethylpiperidin-1-yl)oxyl (TEMPO) derivatives as catholyte to develop high-voltage and stable pH neutral aqueous RFBs.^{19,29,30}

Therefore, the search and development of new water-soluble active materials for improvement of **neutral pH** battery storage systems is increasingly significant and will continue to growth in the future.”

2. I don't find the solubility of the tryptanthrin in the text. This data should be provided and discussed.

Reply from the authors: We thank the referee for pointing this out. The solubility tests were in fact only in the Supplementary Information in Table SI2 (data below). Following the reviewer suggestion, the data is now provided and discussed in the main text of the new version of the manuscript.

The following sentences were incorporated in the new version of the manuscript:

“Solubility measurements showed that TRYP-SO₃H has a solubility of approximately 0.12 M in 1.0 M KCl at 293 K (see Table SI2 in SI). Therefore, the concentration of active materials used is relatively low to avoid the possible precipitation of active materials during the cycling.”

“In order to study the effect of the concentration of the all-organic active materials in the stability and performance of the RFB, the concentration of the redox pair was increased to 0.1 M, a concentration close to the solubility limit of TRYP-SO₃H (0.12 M in 1.0 M KCl, see Table SI2 in SI).”

Data from Table SI2:

Table SI2. Solubility of K₄[Fe(CN)₆].3H₂O, BQDS and TRYP-SO₃H in 1.0 M KCl at T = 293 K.

Active Material	Solubility (M)
K ₄ [Fe(CN) ₆].3H ₂ O	0.812 - 0.787
BQDS	1.280 - 1.185
TRYP-SO ₃ H	0.122 - 0.108

4. In Figure 5, capacity data should be provided and discussed in the main text.

Reply from the authors: The reviewer suggestion was accepted. Capacity data was provided and discussed in the revised manuscript. The paragraph below was changed (highlighted text) accordingly:

“The obtained average discharge energy density and average discharge capacity of the organometallic active materials cell was 0.014 Wh L^{-1} and 1.17 mAh , respectively, while that of the all-organic TRYP-SO₃H/BQDS redox couple are highest, with values of 0.046 Wh L^{-1} and 2.65 mAh . Long-time capacity stability is a vital characteristic for aqueous RFBs. Indeed, while for the organometallic active materials cell (Fig. 4a) during fifty complete cycles ($\sim 7 \text{ h}$) there is a decrease in the charge and discharge energy density and capacity, with capacity retention falling to 38% of its original value (6.309 to 2.409 C) over fifty cycles (see Fig. SI11a in SI). In general, non-capacity-related coulombic efficiency loss (capacity fades over the cycles but coulombic efficiency increases, Fig. SI11a) mainly arises from electrolyte side reactions.⁵⁸ In the all-organic active materials cell (Fig. 4b) the values steadily increase until ~ 10 cycles and further stabilizes to a total number of fifty cycles ($\sim 29 \text{ h}$). Stable capacity retention was observed with more than 98% total capacity retention after forty cycles (Fig SI11b).”

“It can be seen that at neutral pH the aqueous all-organic active materials flow cell presents a notable cycling retention (98%) and coulombic efficiencies above $\sim 90\%$ through several cycles (see Fig. SI11b in SI).”

Figure SI11 was incorporated in SI:

Fig. S11 Discharge capacity and coulombic efficiency vs cycling numbers plots of neutral pH aqueous organometallic and all-organic active materials for RFB single cells using 5.0 mM of TRYP-SO₃H in 1.0 M KCl solution as supporting electrolyte. **a** 10.0 mM of $K_4[Fe(CN)_6] \cdot 3H_2O$ as catholyte. **b** 5.0 mM of BQDS as catholyte.

5. It seems the tryptanthrin AORFBs were tested at low concentrations. Is it possible to try a higher concentration such as 0.5 M? If not, then I suggest to the authors provide NMR and/or other spectroscopic studies to gain possible degradation of the compound.

Reply from the authors: We appreciate the referee comment and suggestion. Considering the solubility of TRYP-SO₃H in 1.0 M KCl at T = 293 K (0.122 - 0.108 M) it was not possible to increase the concentration to 0.5 M since this is a higher concentration than the solubility limit of TRYP-SO₃H.

Following the second referee suggestion, we performed additional experiments in order to provide supplementary information on the degradation of TRYP-SO₃H. The spectroscopic study, as well as the ¹H NMR spectra were performed on the anolyte (TRYP-SO₃H).

As can be seen in Figure X1 below, the UV-visible spectra of the analyte (TRYP-SO₃H in 1.0 M KCl) before (full black line) and after (full red line) are found different. During cycling the TRYP-SO₃H solution has lost part of the original absorption band at ~400 nm.

¹H NMR analysis of the analyte after cycling in D₂O (Figure X2) shows a significant change in the structure of the analyte. To set of peaks are distinguishable, one corresponding to the main degradation product (singlet at 8.73 ppm, duplet at 8.33 ppm, double duplet at 7.88 ppm and duplet at 7.20 ppm) and a set of less intense peaks with more complex pattern. In any case, the number peaks at the region typical of hydrogens at aromatic structure of tryptanthrin (9-7 ppm) decrease. Therefore, either the number of hydrogens decreases or the symmetry of the molecule increases. The structural modification would likely be the product of substitution reactions similar to the previous reported when BQDS was used in aqueous RFBs (ref. Yang, B. *et al. J. Electrochem. Soc.* **163**, A1442-A1449, 2016); however, separation of the mixture of compounds and full structural characterization would be needed to propose the structure of the degradation product and further and detailed studies would be needed. We consider this would imply a delay of this publication and is presently out of the scope of the present work. Nevertheless and as indicated by the referee, wthis result further supports the chemical modification of TRYP-SO₃H after 50 cycles.

Figure X1 - UV-Visible spectra of the analyte (TRYP-SO₃H in 1.0 M KCl) before (black line) and after (red line) ~29 h (50 charge/discharge cycles) diluted for 10⁴ times.

Figure X2 - ¹H NMR spectra of analyte (TRYP-SO₃H) before (orange) and after cycling (blue).